# FIRE ON MOTION: OPTIMIZING VIDEO PASS-BANDS FOR EFFICIENT SPIKING ACTION RECOGNITION

## ABSTRACT

Spiking neural networks (SNNs) have gained traction in vision due to their energy efficiency, bio-plausibility, and inherent temporal processing. Yet, despite this temporal capacity, most progress concentrates on static image benchmarks, and SNNs still underperform on dynamic video tasks compared to artificial neural networks (ANNs). In this study, we identify a fundamental issue related to pass-band mismatch in SNNs. Typically, standard spiking dynamics function as temporal low-pass filters. This means they tend to highlight static content while diminishing the importance of motion-related frequency bands. However, these motion-bearing bands often contain crucial task-relevant information, especially in dynamic tasks. This phenomenon sheds light on why SNNs can perform comparably to ANNs on static tasks, yet often lag behind when it comes to tasks requiring a deeper temporal understanding. To address this challenge, we propose the Pass-Bands Optimizer (PBO), a plug-and-play module that optimizes the temporal pass-band toward task-relevant motion bands. It introduces only two learnable parameters, and a lightweight consistency constraint that preserves semantics and boundaries, incurring negligible computational overhead and requires no architectural changes. The proposed PBO deliberately suppresses static components that contribute little to discrimination, effectively high-passing the stream so that spiking activity concentrates on motion bearing content. On UCF101, PBO yields over 10% improvement. On more complex multi-modal action recognition and video anomaly detection tasks, PBO delivers consistent and significant gains, offering a new perspective for SNN based video processing and understanding.

## 1 INTRODUCTION

Spiking neural networks (SNNs), the third generation of neural networks (Maass, 1997), have attracted growing interest for their event-driven computation, biological plausibility, and energy efficiency (Akopyan et al., 2015). Unlike continuously active artificial neural networks (ANNs), SNNs maintain a temporal state that integrates inputs and emit spikes only upon crossing a threshold (Gerstner et al., 2014), with activation effectively encoded by spike *rate or timing* rather than fixed nonlinearities (*e.g.,* ReLU). However, this temporal machinery remains under-exploited. Most empirical progress has focused on static vision tasks (*e.g.,* image classification), often creating a pseudo-temporal dimension by replicating a single frame. On such benchmarks, recent SNN models can match or even exceed strong ANN counterparts while preserving attractive sparsity and low latency Zhou et al. (2024). Yet when tasks genuinely rely on *temporal* reasoning and motion cues, such as action recognition (Ahmad et al., 2021; Jhuang et al., 2013), video anomaly detection (VAD) (Qian et al., 2025), and broader video understanding, SNN performance still falls short of expectations.

Empirically, in RGB video streams, abundant static background and low-frequency redundancy elicit large volumes of motion-irrelevant spikes, consuming a limited spiking budget. Previous work has noted that SNNs can benefit from residual inputs relative to RGB (Xiao et al., 2024), suggesting that sparse, motion-dominant signals better match spiking computation. However, pure first-order differencing removes DC entirely: it improves sparsity but discards substantial semantic content, making the sparsity–semantics trade-off difficult to measure and optimize. Meanwhile, analysis shows that SNNs tend to attenuate high-frequency content (Fang et al., 2025), while they addresses this deficiency by boosting *spatial* high frequencies (*e.g.,* max-pooling) but it does not directly restore the *temporal* bands that encode motion, and evaluations largely remain on static benchmarks.

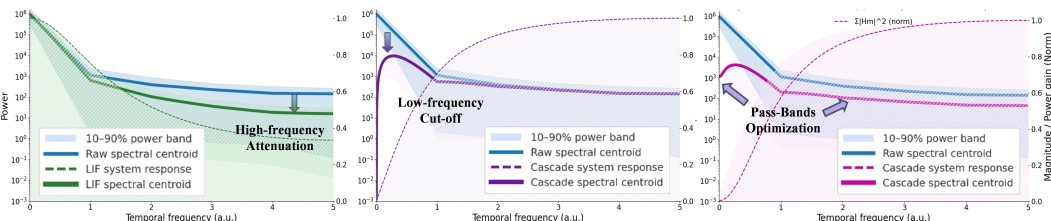

Figure 1: **Temporal power spectra computed over the full UCF101 dataset** (Soomro et al., 2012) and the effects of different filters. (a) The LIF dynamics act as a low-pass filter, suppressing high-frequency components. (b) Cascading a temporal high-pass with the LIF stage retains high-frequency content but eliminates low-frequency energy, nulling the DC component. (c) Our plug-and-play module with LIF adaptively optimizes the temporal pass-band in a task-driven manner, yielding a task-optimal pass-band.

To investigate, we revisit SNNs from a frequency-domain perspective and offer a unified diagnosis: a *pass-band mismatch*. In SNN models, the LIF neuron's subthreshold membrane voltage is equivalent to a linear **low-pass transformation** of the temporal input (Naud & Gerstner, 2012), which preserves direct current (DC) and ultra-low-frequency content while attenuating nonzero temporal components. In contrast, discriminative information in natural videos concentrates in nonzero, **mid-frequency motion bands**. As shown in Fig. 1 (a), raw videos concentrate energy at DC, whereas task-relevant motion lies in the mid bands. Standard SNN processing compounds attenuation at these bands and mainly retains DC/low-frequency content. In Fig. 1 (b), cascading a first-order temporal high-pass hard-truncates the spectrum at $\omega \approx 0$: it zeros the DC component and strongly attenuates low frequencies, causing a complete loss of low-frequency and steady-state information.

Therefore, neither raw-frame input nor simple first-order differencing can realize a task-optimal pass-band, as their pass-bands are fixed. To resolve this, we introduce the **Pass-Band Optimizer** (PBO): a plug-and-play, extremely simple, causal, *linear* prefilter that *optimizes the input pass-band in a data-driven manner prior to membrane integration*, reshaping the overall temporal response from low-pass toward a task-aligned band-pass. Extensive experiments show significant and consistent gains across *uni-modal* settings (RGB), *multi-modal* fusion (RGB + event streams), and *weakly supervised video anomaly detection*. In Fig. 1 (c), PBO suppresses DC/near-DC energy to the level of mid/high frequencies, with only two learnable scalars before the embedding stage and no extra training/inference cost. In RGB-event fusion (UCF101 + UCF101-DVS), PBO raises accuracy from 68.13% to 73.03%. PBO preserves sparsity and low latency, and is compatible with spatial enhancement modules, and learnable neuron time constants. We view the SNN-on-video bottleneck as a temporal *pass-band mismatch* and offer a minimal, architecture-agnostic, streaming-friendly, plug-and-play module for existing SNNs that strengthens temporal modeling on truly dynamic tasks.

**Contributions.** (1) We are the first to diagnose and analyze the *temporal pass-band mismatch* in SNN video processing, providing a frequency-domain perspective for further SNN-based video understanding. (2) We propose *Pass-Band Optimizer* (PBO), a plug-and-play causal pre-filter inserted *before* the membrane with only *two* learnable scalars. PBO reshapes the response from low-pass to task-aligned pass-band without changing backbones. (3) We introduce a consistency constraint for Pass-Band Optimization and validate its effectiveness on three tasks that require motion understanding: uni-modal (RGB) action recognition, multi-modal (RGB+DVS) action recognition, and weakly supervised video anomaly detection. Experimental results demonstrate that PBO achieves stable and significant performance gains, showcasing robust generalization on dynamic tasks.

## 2 PRELIMINARY AND RELATED WORK

**Spiking neural networks (SNNs)** replace continuous activation function (*e.g.,* ReLU) with spike neurons, enabling spike-driven sparsity and temporal state via membrane dynamics and resets. In this work, we adopt the widely used discrete-time Leaky Integrate-and-Fire (LIF) neuron (Gerstner et al., 2014). The membrane potential and spike firing of the LIF model are governed by

$$U[t] = V[t-1] + \tau\left(X[t] - (V[t-1] - V_{\text{reset}})\right),\ S[t] = \Theta(U[t] - V_{\text{th}}),\ V[t] = U[t]\left(1 - S[t]\right) + V_{\text{reset}}S[t],\ (1)$$

where $\tau \in (0, 1)$ is the leak coefficient; $X[t]$ and $V[t]$ denote the input and membrane potential at time step $t$. $U[t]$ is the pre-spike membrane potential while $S[t] \in \{0, 1\}^d$ is the binary spike computed via the Heaviside function $\Theta(\cdot)$ with firing threshold $V_{\text{th}}$ and the reset potential $V_{\text{reset}}$.

**SNN-based Discrete-Time Frequency Analysis.** SNNs possess unique temporal modeling capacity for dynamic vision, and several works have analyzed their frequency characteristics. FSTA-SNN (Yu et al., 2025) reports strong temporal redundancy across time steps and introduces a frequency-based spatiotemporal attention to suppress it. Max-Former (Fang et al., 2025) argues SNNs behave as low-pass at the network level and restores missing high-frequency content via extra max-pooling and early depth-wise convolution. However, these approaches remain largely confined to image classification (or its DVS counterpart), where task signals are dominated by static appearance. For RGB video understanding, SNN analysis and methods are still scarce; thus we analyze and optimize temporal pass-bands in the frequency domain.

With the above context, we collect the discrete-time frequency preliminaries and notation for our subsequent analysis. Given a discrete-time sequence $x[t]$ of length $T$, we work directly with its Discrete-Time Frequency Transformation (DTFT): $X(e^{j\omega}) = \sum_{t=0}^{T-1} x[t]\, e^{-j\omega t}$, where $j^2 = -1$ and $\omega \in [-\pi, \pi]$ is the normalized angular frequency in radians per sample. Noting that $\omega = 0$ indicates DC and $\omega = \pi$ is the Nyquist edge. Let $h[t]$ be an impulse response with frequency response $H(e^{j\omega})$. For a linear time-invariant (LTI) system, its output $y[t]$ is the time-domain convolution of $h[t]$ and $x[t]$, corresponding to the frequency-domain multiplication:

$$y[t] = h[t] * x[t] = \sum_{\tau=-\infty}^{\infty} h[\tau]\, x[t-\tau] \overset{\text{DTFT}}{\longleftrightarrow} Y(e^{j\omega}) = H(e^{j\omega})\, X(e^{j\omega}). \tag{2}$$

For low-frequency redundancy or contamination, a low-pass stage $H_{\text{LP}}(\cdot, \omega_c)$ with cutoff $\omega_c \in (0, \pi)$ preserves baseband and attenuates the high end, with ideal magnitude:

$$|H_{\text{LP}}(e^{j\omega}, \omega_c)| = \mathbb{I}\{\, |\omega| \le \omega_c\, \}, \quad where \quad |H_{\text{LP}}(e^{j\pi}, \omega_c)| = 0, \tag{3}$$

where $\mathbb{I}$ is the indicator. Conversely, when high-frequency noise dominates, a high-pass stage $H_{\text{HP}}(\cdot, \omega_c)$ with the same cutoff suppresses DC and retains the high end:

$$|H_{\text{HP}}(e^{j\omega}, \omega_c)| = \mathbb{I}\{\, |\omega| \ge \omega_c\, \}, \quad where \quad |H_{\text{HP}}(e^{j0}, \omega_c)| = 0. \tag{4}$$

Cascading these two yields a band-pass filter $H_{\text{BP}}(\cdot, \omega_1, \omega_2)$ with $0 < \omega_1 < \omega_2 \le \pi$:

$$H_{\text{BP}}(e^{j\omega}, \omega_1, \omega_2) = H_{\text{HP}}(e^{j\omega}, \omega_1)\, H_{\text{LP}}(e^{j\omega}, \omega_2), \quad |H_{\text{BP}}(e^{j\omega})| = \mathbb{I}\{\, \omega_1 \le |\omega| \le \omega_2\, \}. \tag{5}$$

Subsequently, we use these transforms and their filter relations, treating membrane integration, leakage and synaptic dynamics as frequency-shaping filters, to analyze dynamic-vision data and learn pass-bands aligned with task-relevant temporal structure.

## 3 ANALYSIS OF PASS-BANDS MISMATCH UNDER LIF CONSTRAINTS

In this work, we model the input to a spiking layer as a discrete-time vector sequence $\boldsymbol{X}[t] \in \mathbb{R}^d$ of length $T$, which can be decomposed as:

$$\boldsymbol{X}[t] = \boldsymbol{B} + \boldsymbol{M}[t] + \boldsymbol{n}[t], \tag{6}$$

where $\boldsymbol{B} \in \mathbb{R}^d$ concentrates at DC and ultra-low frequencies, $\boldsymbol{M}[t]$ captures action-induced dynamics with the angular frequency $\omega > 0$, and $\boldsymbol{n}[t]$ is additive noise. At the *subthreshold* steps in a LIF neuron defined by Eq. 1, the binary spike vector $S[t] = 0$, i.e. $V[t] = U[t]$, which leads to:

$$U[t] = V[t-1] + \tau\Big(X[t] - \big(V[t-1] - V_{\text{reset}}\big)\Big) = (1-\tau)\,V[t-1] + \tau V_{\text{reset}} + \tau X[t]. \tag{7}$$

Then, recentring $\tilde{\boldsymbol{V}}[t] \triangleq \boldsymbol{V}[t] - V_{\text{reset}}$ yields the first-order linear recursion:

$$\tilde{\boldsymbol{V}}[t] = \alpha\, \tilde{\boldsymbol{V}}[t-1] + (1-\alpha)\, \boldsymbol{X}[t], \quad where \quad \alpha \triangleq 1 - \tau \in (0, 1). \tag{8}$$

Taking the DTFT over $t$ gives the temporal frequency response:

$$H_{\text{LIF}}(e^{j\omega}) = \frac{\tilde{V}(e^{j\omega})}{X(e^{j\omega})} = \frac{1-\alpha}{1 - \alpha e^{-j\omega}}, \quad with \quad \big|H_{\text{LIF}}(e^{j\omega})\big|^2 = \frac{(1-\alpha)^2}{1 + \alpha^2 - 2\alpha\cos\omega}, \tag{9}$$

which is a classic **temporal low-pass** with passband near $\omega = 0$ and increasing attenuation as $\omega$ grows. Let $S(\omega)$ denote the DTFT-based power spectral density (PSD) of a wide-sense stationary input. Since $X[t]$ admits the linear decomposition in Eq. 6, the PSD of the LTI input $S_{\text{in}}(\omega)$ can be decomposed into the energies of corresponding components:

$$S_{\text{in}}(\omega) \equiv S_X(\omega) = S_B(\omega) + S_M(\omega) + S_n(\omega). \tag{10}$$

Then, its output $S_{\text{out}}(\omega)$ gives as:

$$S_{\text{out}}(\omega) \;=\; \big|H_{\text{LIF}}(e^{j\omega})\big|^2 S_{\text{in}}(\omega) \;=\; \big|H_{\text{LIF}}(e^{j\omega})\big|^2 \Big(S_B(\omega) + S_M(\omega) + S_n(\omega)\Big). \tag{11}$$

Since $|H_{\text{LIF}}(e^{j0})|^2 = 1$, $S_B(\omega) = S_B(0)$, and $S_M(0) = 0$ because motion resides at $\omega > 0$, DC pass essentially unattenuated while motion is suppressed at nonzero bands:

$$S_{\text{out}}(0) \;=\; \big|H_{\text{LIF}}(e^{j0})\big|^2 \big(S_B(0) + S_n(0)\big) \;+\; \big|H_{\text{LIF}}(e^{j0})\big|^2 S_M(0) \;=\; S_B(0) + S_n(0). \tag{12}$$

For any fixed $\omega_0 > 0$, low-pass attenuation implies

$$\int_{\omega_0}^{\pi} \big|H_{\text{LIF}}(e^{j\omega})\big|^2 S_M(\omega)\, d\omega \;\leq\; \varepsilon(\alpha, \omega_0) \int_{\omega_0}^{\pi} S_M(\omega)\, d\omega, \quad \varepsilon(\alpha, \omega_0) \triangleq \max_{\omega \in [\omega_0, \pi]} \big|H_{\text{LIF}}(e^{j\omega})\big|^2 \ll 1, \tag{13}$$

which formalizes that $B$ and low-frequency noise consume spike budget, while the motion-bearing (task-relevant) component $M[t]$ is heavily attenuated after the membrane. This explains why SNNs can achieve comparable performance to ANNs on static image tasks, yet struggle on dynamic tasks due to the significant loss of motion information. To be precise, without explicit frequency-domain processing, SNNs primarily rely on the DC and ultra-low-frequency components of the video. We refer to this phenomenon as **pass-band mismatch**. To address this issue, we aim to design and cascade a **learnable pre-filter** $H(e^{j\omega}; \theta)$ with the frequency coefficient $\theta$ before the membrane:

$$S_{\text{out}}^{(\theta)}(\omega) \;=\; \big|H(e^{j\omega}; \theta)\big|^2 \big|H_{\text{LIF}}(e^{j\omega})\big|^2 S_{\text{in}}(\omega). \tag{14}$$

Let $\mathcal{G}(\cdot; \theta)$ denotes the model that cascades $H(e^{j\omega}; \theta)$ with the membrane and the task head. Given $N$ pairs of input $X_i[t]$ and label $y_i$, and supervised loss $\ell$ (*e.g.,* cross-entropy), the optimization is:

$$\min_{\theta} \; \mathcal{L}(\theta) = \frac{1}{N} \sum_{i=1}^{N} \ell(\mathcal{G}(X_i[t]; \theta), y_i). \tag{15}$$

## 4 METHODOLOGY

**Motivation.** Despite the temporal modeling capacity of LIF neurons, our analysis (Sec. 3) shows that the LIF constraint induces a temporal low-pass whose pass-band is **mismatched** with video dynamics. We therefore insert a *learnable, causal* pre-filter *before* the embedding stack and optimize it during training to obtain a **task-aligned pass-band**. In addition, we introduce a *consistent loss* to find a dynamic balance between low-pass and high-pass systems.

### 4.1 TEMPORAL PRE-FILTER AND CASCADED RESPONSE

**Definition.** We first define the pass-bands pre-filter with a two-point shift-and-subtract:

$$\boldsymbol{Y}^{(\lambda)}[t] \;=\; \boldsymbol{X}[t] - \lambda\, \boldsymbol{X}[t-1] \;=\; (1-\lambda)\, \boldsymbol{X}[t] + \lambda\big(\boldsymbol{X}[t] - \boldsymbol{X}[t-1]\big), \quad \lambda \in [0, 1]. \tag{16}$$

This operation can be expressed as a learnable weighted sum of the original frame and the frame difference, making it minimal and lightweight. Its frequency response is

$$W(e^{j\omega}, \lambda) = 1 - \lambda e^{-j\omega}, \quad with \quad \big|W(e^{j\omega}, \lambda)\big|^2 = 1 + \lambda^2 - 2\lambda \cos\omega. \tag{17}$$

**Cascaded Frequency Response.** Using the LIF temporal frequency response in Eq. 9, the cascaded transfer becomes

$$G(e^{j\omega}, \lambda) = W(e^{j\omega}, \lambda)\, H_{\text{LIF}}(e^{j\omega}) = \frac{(1 - \lambda e^{-j\omega})(1 - \alpha)}{1 - \alpha e^{-j\omega}}, \quad \big|G(e^{j\omega}, \lambda)\big|^2 = \frac{(1 + \lambda^2 - 2\lambda \cos\omega)(1 - \alpha)^2}{1 + \alpha^2 - 2\alpha \cos\omega}. \tag{18}$$

However, with fixed $\alpha$, Eq. 18 implies that as $\lambda$ sweeps $0 \to 1$, the DC gain $|G(e^{j0})|^2 = (1 - \lambda)^2$ decreases while the high-frequency endpoint $|G(e^{j\pi})|^2 = (1 + \lambda)^2(1 - \alpha)^2(1 + \alpha)^{-2}$ increases. The response is monotone in $\omega$, i.e. low-pass tilt if $\lambda < \alpha$, flat at $\lambda = \alpha$ and high-pass tilt if $\lambda > \alpha$. Thus, a single $\lambda$ only shifts the passband centroid and cannot form a mid-band peak or independently control bandwidth (more details in Appendix A). We therefore generalize to a **time-varying** $\lambda[t]$.

### 4.2 PASS-BANDS OPTIMIZER

**LTV pass-bands optimizer definition.** To enlarge the optimizable pass-band, in both shape and spectral center, before the LIF layer, we generalize the scalar $\lambda$ to a **time-varying sequence** $\lambda[t]$, yielding a two-tap linear time-varying (LTV) pre-filter. The formulation in Eq. 16 is thus redefined:

$$\boldsymbol{Y}[t] \;=\; \boldsymbol{X}[t] - \lambda[t]\boldsymbol{X}[t-1], \quad where \quad h[t, 0] = 1, \; h[t, 1] = -\lambda[t], \; h[t, k] = 0 \; (k \notin \{0, 1\}). \tag{19}$$

**A plug-and-play stationarity-aware periodic pre-filter.** To maintain architectural simplicity and support seamless deployment, such LTV pass-bands optimizer is designed as a plug-and-play module to generate a time-varying coefficient sequence $\lambda[t]$ before the first spiking membrane, without modifying the backbone or inference flow. To respect spectral stationarity and enable interpretable frequency-domain shaping, we adopt a *bounded-energy, mean-stable, cyclostationary* parameterization of $\lambda[t]$ determined by two learnable parameters $\mu$ and $\omega$, which preserves a well-defined DC baseline with controllable frequency-sideband structures. Concretely, we set

$$\lambda[t] \;=\; \mu \;+\; A\,\sin(\omega\,t + \phi), \quad \text{where} \quad \mu \in [0,1], \quad A \geq 0, \quad \omega \in (0,\pi], \quad \phi \in \mathbb{R}. \tag{20}$$

Here, $\mu$ determines the time-average (DC component) of $\lambda[t]$, which governs the mean behavior of the pre-filter (*e.g.,* stronger DC suppression as $\mu$ increases). Meanwhile, the sinusoidal modulation introduces structured nonzero-frequency components that broaden and shape the effective pass-band.

**Parameterization and initialization.** Given the learnable mean $\mu \in [0,1]$ and angular frequency $\omega \in (0,\pi)$, $\omega$ is indirectly determined by a learnable raw variable $\sigma_{\text{raw}} \in \mathbb{R}$ via a logistic map: $p = \sigma(\sigma_{\text{raw}}) = \frac{1}{1+e^{-\sigma_{\text{raw}}}}$, $\omega = \pi\,p$. The mean $\mu$ determines the DC baseline and averages pass-band tilt, initialized at $\mu = 0.5$. It is then guided toward a dynamic equilibrium within $[0,1]$ via a consistency loss defined later in Eq. 27. For default initialization over a clip of length $T$, we target one period via $\omega_0 = \frac{2\pi}{T-1}$ and set $\sigma_{\text{raw}} \leftarrow \log \frac{2/(T-1)}{1-2/(T-1)}$.

**Harmonic-transfer view and dominant sidebands.** Under a $P$-periodic $\lambda[t]$ with fundamental $\omega_0 = 2\pi/P$, the Linear Periodically Time-Varying system (LPTV) response admits the harmonic-transfer representation:

$$Y(e^{j\omega}) \;=\; \sum_{m \in \mathbb{Z}} W_m(e^{j\omega})\,X\big(e^{j(\omega - m\omega_0)}\big), \tag{21}$$

where $\{W_m\}$ are determined by the Fourier coefficients of $\lambda[t]$. For the single-tone model in Eq. 20 with $\phi = 0$ and $\omega = \omega_0$, we have:

$$\lambda_0 = \mu,\ \lambda_{\pm 1} = \mp\frac{A}{2j},\ \lambda_{|m|>1} = 0 \ \Rightarrow\ W_0(e^{j\omega}) = 1 - \mu\,e^{-j\omega},\ W_{\pm 1}(e^{j\omega}) = -\lambda_{\pm 1}\,e^{-j\omega},\ W_{|m|>1}(e^{j\omega}) = 0. \tag{22}$$

**Cascade with LIF and PSD approximation.** By the response in Eq. 9, the cascaded spectrum is:

$$Y(e^{j\omega}) \;=\; H_{\text{LIF}}(e^{j\omega}) \sum_{m \in \mathbb{Z}} W_m(e^{j\omega})\,X\big(e^{j(\omega - m\omega_0)}\big). \tag{23}$$

For convenience of derivation, we assume approximate uncorrelatedness across frequency bins (the full correlated version is derived in Appendix G, and the resulting sideband conclusion remains unchanged). The output PSD is approximated by:

$$S_{\text{out}}(\omega) \;\approx\; \big|H_{\text{LIF}}(e^{j\omega})\big|^2 \Big(\big|W_0(e^{j\omega})\big|^2 S_{\text{in}}(\omega) + \sum_{m \neq 0} \big|W_m(e^{j\omega})\big|^2 S_{\text{in}}(\omega - m\omega_0)\Big). \tag{24}$$

For the single-tone case $\omega = \omega_0$ with $\phi = 0$, it can be simplified as:

$$S_{\text{out}}(\omega) \;\approx\; \big|H_{\text{LIF}}(e^{j\omega})\big|^2 \Big(\underbrace{\big|1 - \mu\,e^{-j\omega}\big|^2 S_{\text{in}}(\omega)}_{\text{baseline (DC) term}} + \underbrace{\tfrac{A^2}{4}\,S_{\text{in}}(\omega - \omega_0) + \tfrac{A^2}{4}\,S_{\text{in}}(\omega + \omega_0)}_{\text{frequency-translation sidebands}}\Big), \tag{25}$$

which reveals that the time variation injects controllable sidebands at $\pm\omega_0$ which **translate low-frequency energy into nonzero bands**. Choosing $\omega_0$ inside the LIF-transmissible region yields a genuine mid-band pass window, while $\mu$ sets the DC floor and hyper-parameter $A$ controls peak height and effective bandwidth.

**Why this is reasonable despite time variation?** Even though $\lambda[t]$ is time-varying, our design remains theoretically sound and practically stable for the following reasons: **(i)** The boundedness $\lambda[t] \in [0,1]$ ensures numerical stability and preserves the physical interpretability of the two-tap pre-emphasis filter. **(ii)** From an expectation perspective, the time-varying coefficient sequence behaves equivalently to a constant filter with $\lambda = \mu$, thereby retaining the original high-pass tilt and DC suppression characteristics of the static formulation. Detailed derivation is in Appendix E. **(iii)** Since $\lambda[t]$ is a structured periodic function, it induces a cyclostationary filtering regime that legitimizes the

use of harmonic-transfer analysis in Eq. 21–25, providing a principled mechanism to broaden and shape the effective passband while remaining streaming and computationally lightweight.

**Consistency loss.** To keep the optimized pre-LIF signal faithful to the original semantics, we regularize the learned mixture against the two endpoints in Eq. 16. Let $\boldsymbol{Y}^{(0)}[t] = \boldsymbol{X}[t]$ (DC) and $\boldsymbol{Y}^{(1)}[t] = \boldsymbol{X}[t] - \boldsymbol{X}[t-1]$ (High-frequency), and denote the filtered output by $\boldsymbol{Y}_t^{(m)}$. A weight $\lambda[t] \in [0,1]$ balances the two references. To avoid bias toward either endpoint caused by absolute intensity scale, we *spatially de-mean* the signals in the intensity term. Let

$$\tilde{\boldsymbol{Y}}_t^{(k)} \ = \ \boldsymbol{Y}_t^{(k)} - \frac{1}{HW} \sum_{x=1}^{W} \sum_{y=1}^{H} \boldsymbol{Y}_t^{(k)}(x,y), \quad k \in \{0,1,m\}, \tag{26}$$

computed per time step and per channel. The consistency loss can be formulated as,

$$\mathcal{L}_{\text{consist}} = \frac{1}{TCHW} \sum_{t=0}^{T-1} \left( \mathcal{L}_t^{\text{int}} + \mathcal{L}_t^{\text{grad}} \right),$$

$$\mathcal{L}_t^{\text{int}} = \left\| \lambda[t] \odot \left( \tilde{\boldsymbol{Y}}_t^{(m)} - \tilde{\boldsymbol{Y}}_t^{(1)} \right) \right\|_2^2 + \left\| (1 - \lambda[t]) \odot \left( \tilde{\boldsymbol{Y}}_t^{(m)} - \tilde{\boldsymbol{Y}}_t^{(0)} \right) \right\|_2^2, \tag{27}$$

$$\mathcal{L}_t^{\text{grad}} = \left\| \nabla_x \boldsymbol{Y}_t^{(m)} - \max(|\nabla_x \boldsymbol{Y}_t^{(0)}|, |\nabla_x \boldsymbol{Y}_t^{(1)}|) \right\|_1 + \left\| \nabla_y \boldsymbol{Y}_t^{(m)} - \max(|\nabla_y \boldsymbol{Y}_t^{(0)}|, |\nabla_y \boldsymbol{Y}_t^{(1)}|) \right\|_1$$

where $\nabla_x, \nabla_y$ are Sobel gradients along horizontal and vertical directions. The $L_2$ term penalizes per-pixel deviations from the *demeaned* endpoints according to $\lambda[t]$, thereby preventing intensity-scale attraction to either side and yielding a balanced objective. The $L_1$ term aligns edges by matching the stronger response from either endpoint, preserving sharpness and structural sparsity.

**Closed-form equilibrium of the intensity term.** For fixed $\lambda[t]$, minimizing $\mathcal{L}_t^{\text{int}}$ over $\tilde{\boldsymbol{Y}}_t^{(m)}$ admits a per-pixel closed form:

$$\hat{\tilde{\boldsymbol{Y}}}_t^{(m)} = \arg \min_{\tilde{\boldsymbol{Y}}} \left[ \lambda[t]^2 \left\| \tilde{\boldsymbol{Y}} - \tilde{\boldsymbol{Y}}_t^{(1)} \right\|_2^2 + (1 - \lambda[t])^2 \left\| \tilde{\boldsymbol{Y}} - \tilde{\boldsymbol{Y}}_t^{(0)} \right\|_2^2 \right] = \frac{\lambda[t]^2 \tilde{\boldsymbol{Y}}_t^{(1)} + (1 - \lambda[t])^2 \tilde{\boldsymbol{Y}}_t^{(0)}}{\lambda[t]^2 + (1 - \lambda[t])^2}, \tag{28}$$

which interpolates between the low-pass ($\lambda=0$, $\hat{\tilde{\boldsymbol{Y}}}_t^{(m)} = \tilde{\boldsymbol{Y}}_t^{(0)}$) and the high-pass ($\lambda=1$, $\hat{\tilde{\boldsymbol{Y}}}_t^{(m)} = \tilde{\boldsymbol{Y}}_t^{(1)}$), establishing a well-posed dynamic equilibrium within $[0,1]$. Overall, we train the model and the pass-bands optimizer with a classification loss and two auxiliary terms $\mathcal{L} = \mathcal{L}_{\text{ce}} + \alpha \, \mathcal{L}_{\text{consist}}$.

## 5 EXPERIMENTS

**Datasets and Tasks.** We evaluate three kinds of dynamic vision tasks: uni-modal action recognition, multi-modal action recognition, and video anomaly detection. All experiments use *color and event paired* (CEP) datasets in which RGB and dynamic vision sensor (DVS) streams are spatio-temporally aligned. Since DVS reports per-pixel intensity changes asynchronously with microsecond temporal resolution and sparse activity, the corresponding stream is motion dominant.

**1) UCF101-CEP.** UCF101 (Soomro et al., 2012) has 13,320 RGB videos over 101 classes. Its DVS counterpart, UCF101-DVS (Bi et al., 2020), is generated with the DAVIS240 simulator. The public DVS release contains 13,523 clips, which exceeds the RGB set. Thus, we remove redundancies to enforce one-to-one pairing. Since about half of the DVS clips are horizontally flipped relative to RGB, we flip RGB ones to match the orientation. We refer to the aligned pair set as **UCF101-CEP**.

**2) HMDB51-CEP.** HMDB51 (Kuehne et al., 2011) has 6,766 RGB clips in 51 classes. Paired with its DVS counterpart (Bi et al., 2020) generated with the DAVIS240, referred as **HMDB51-CEP**.

**3) HARDVS.** HARDVS (Wang et al., 2024) is the largest DVS action recognition dataset recorded with DAVIS346, containing >100,000 clips over 300 classes. It offers naturally captured, temporally aligned RGB-DVS streams and is challenging due to scale, diversity, and realistic conditions.

**4) UCF-Crime-CEP.** UCF-Crime and UCF-Crime-DVS (Qian et al., 2025) form the largest VAD set with RGB and event modalities. We use both inputs and compare against representative ANN and SNN baselines on this fine-grained anomaly detection task.

**Implementation Details.** Experiments are conducted on the BrainCog platform (Zeng et al., 2023) using four NVIDIA RTX 4090 GPUs. We train all models with AdamW (initial learning rate 0.005).

Table 1: Results of different backbones with vs. without PBO on UCF101 and HMDB51.

| Dataset | Methods | Architecture | Params | $T$ | Acc(%) | $\Delta$(%) |
|---|---|---|---|---|---|---|
| UCF101 | Spikformer (Zhou et al., 2023b) | Spikformer-2-256 | 2.58M | 10 | 46.16[*] | – |
| | **Spikformer + PBO** | **Spikformer-2-256** | **2.58M** | **10** | **57.71** | **+11.55** |
| | SDT-V1 (Yao et al., 2023a) | SD-Transformer-2-256 | 2.59M | 10 | 49.25[*] | – |
| | **SDT-V1 + PBO** | **SD-Transformer-2-256** | **2.59M** | **10** | **59.80** | **+10.55** |
| HMDB51 | Spikformer (Zhou et al., 2023b) | Spikformer-2-256 | 2.58M | 10 | 58.66[*] | – |
| | **Spikformer + PBO** | **Spikformer-2-256** | **2.58M** | **10** | **65.22** | **+6.56** |
| | SDT-V1 (Yao et al., 2023a) | SD-Transformer-2-256 | 2.59M | 10 | 62.24[*] | – |
| | **SDT-V1 + PBO** | **SD-Transformer-2-256** | **2.59M** | **10** | **68.21** | **+5.97** |

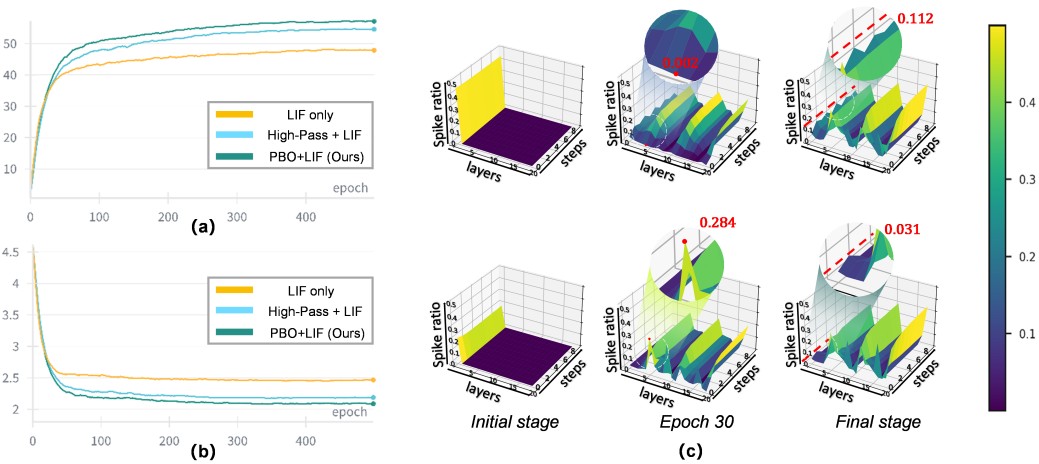

Figure 2: UCF101 results with a spike-driven transformer (Yao et al., 2023a). (a) Top-1 accuracy vs. epoch for three schemes: LIF only (low-pass), High-pass → LIF (coarse band-pass), and PBO → LIF (ours). (b) Corresponding validation loss. (c) Layer-step spike-ratio surfaces of the LIF across training. Left→right: initial, epoch 30, final. Top row: LIF only, bottom row: PBO→LIF. Color encodes spike ratio in $[0, 0.5]$. *Red dots indicate the firing ratio used for query mapping.*

Unless stated otherwise, the LIF node uses a time constant $\tau = 0.7$ and a firing threshold of 1, the amplitude $A$ and the phase $\phi$ are set to 0.1 on all the datasets. To assess the plug-and-play nature of PBO, we compare against baselines taken from the strongest numbers reported in the original papers when available. Otherwise, we reimplement the methods under their stated settings and use the reproduced accuracy as the baseline. We then insert PBO into the same backbones without changing the architecture, loss, data preprocessing, training schedule, or compute budget, and we report improvements under the same sequence length and input resolution.

### 5.1 MAIN RESULTS ON UNI-MODAL AND MULTI-MODAL ACTION RECOGNITION

**Uni-modal effectiveness on RGB video datasets.** As shown in Table 1, our plug-and-play Pass-Band Optimizer (PBO) brings large gains on UCF101 and HMDB51. On UCF101, PBO lifts Spikformer from **46.16% to 57.71%** and SDT-V1 from **49.25% to 59.80%**. On HMDB51, we observe similar consistent improvements: **58.66% to 65.22%** for Spikformer and **62.24% to 68.21%** for SDT-V1. These gains arrive without altering the backbone or inference cost, underscoring that aligning a models temporal pass-band with task-relevant motion bands is a first-order factor for RGB-based action recognition in SNNs.

From the learning dynamics presented in Fig. 2 (a) and (b), PBO reaches *lower* validation loss than either (i) a coarse band-pass formed by naively cascading High-pass → LIF, or (ii) the LIF-only low-pass. While convergence speed in epochs is comparable, the final loss plateau of our PBO is markedly smaller, indicating a better optimal solution rather than a mere optimization acceleration.

The spike-ratio surfaces given in Fig. 2 (c) also support our motivation and theory. By **epoch 30**, PBO has already activated the $Q_{\text{LIF}}$ pathway within the spiking self-attention (SSA) (Zhou et al., 2023a), producing structured layer-step selective firing (particularly in mid-temporal steps), whereas

Table 2: Comparison with existing methods on UCF101-DVS, HMDB51-DVS, and HARDVS datasets. * Results reproduced under our unified implementation framework.

| Dataset | Category | Methods | Architecture | Params | $T$ | Accuracy |
|---|---|---|---|---|---|---|
| UCF101-DVS | ANN | 3D CNN (Tran et al., 2015) | C3D | 78.41M | 8/16 | 38.2 / 47.2 |
| | | RG-CNN (Bi et al., 2020) | RG-CNN + Incep. 3D | 6.95M | 8/16 | 63.2 / 67.8 |
| | | ESCNet (Chen et al., 2022) | ESCNet-SES | – | 8/16 | 59.9 / 70.2 |
| | SNN | RM-SNN (Yao et al., 2023b) | ResNet-18 | – | 8 | 58.5 |
| | | TIM (Shen et al., 2024) | Spikformer-2-256 | 2.58M | 10 | 63.8 |
| | | TIM (Shen et al., 2024) | SD-Transformer-2-256 | 2.59M | 10 | 64.38* |
| | Multi-modal SNN | SCA (Guo et al., 2023) | Spikformer-2-256 | 3.60M | 10 | 60.11* |
| | | WeiAttn (Liu et al., 2022) | SD-Transformer-2-256 | 3.33M | 10 | 67.58* |
| | | CMCI (Jiang et al., 2023) | SD-Transformer-2-256 | 4.44M | 10 | 65.69* |
| | | S-CMRL (He et al., 2025) | SD-Transformer-2-256 | 4.10M | 10 | 68.13* |
| | | **S-CMRL + PBO (Ours)** | **SD-Transformer-2-256** | **4.10M** | **10** | **73.03** |
| HMDB51-DVS | ANN | 3D CNN (Tran et al., 2015) | C3D | 78.41M | 8/16 | 34.2 / 41.7 |
| | | RG-CNN (Bi et al., 2020) | RG-CNN + Incep. 3D | 6.95M | 8/16 | 45.2 / 51.5 |
| | | I3D (Carreira & Zisserman, 2017) | I3D | 12.37M | 8/16 | 38.6 / 46.6 |
| | SNN | RM-SNN (Yao et al., 2023b) | ResNet-18 | – | 8 | 44.7 |
| | | TIM (Shen et al., 2024) | Spikformer-2-256 | 2.58M | 10 | 58.6 |
| | | TIM (Shen et al., 2024) | SD-Transformer-2-256 | 2.59M | 10 | 61.93* |
| | Multi-modal SNN | SCA (Guo et al., 2023) | Spikformer-2-256 | 3.60M | 10 | 70.15* |
| | | WeiAttn (Liu et al., 2022) | SD-Transformer-2-256 | 3.32M | 10 | 71.94* |
| | | CMCI (Jiang et al., 2023) | SD-Transformer-2-256 | 4.40M | 10 | 71.64* |
| | | S-CMRL (He et al., 2025) | SD-Transformer-2-256 | 4.09M | 10 | 72.33* |
| | | **S-CMRL + PBO (Ours)** | **SD-Transformer-2-256** | **4.09M** | **10** | **74.18** |

Table 3: Comparison of accuracy and energy consumption on HARDVS.

| Type | Method | Model | Params | $T$ | Energy (mJ) | $\Delta$ (%) | Acc (%) |
|---|---|---|---|---|---|---|---|
| ANN | ACTION-Net (Wang et al., 2021) | ResNet-50 | 27.9M | 8 | – | – | 46.85 |
| | TimeSformer (Bertasius et al., 2021) | ViT-B/16 | 121.2M | 8 | – | – | 50.77 |
| | ESTF (Wang et al., 2024) | ResNet-18 | 46.7M | 8 | 81.1 | – | 51.2 |
| | TSM (Lin et al., 2019) | ResNet-50 | – | 8 | 87.4 | – | 52.6 |
| SNN | SDT-V1 (Yao et al., 2023a) | SDT-V1 | 2.6M | 8 | – | – | 36.5 |
| | SDT-V2 (Yao et al., 2024) | SDT-V2 | 18.3M | 8 | 8.0 | – | 47.5 |
| | SDT-V3 (Yao et al., 2025) | SDT-V3 | 18.7M | 8 | 23.5 | – | 49.2 |
| Multi- modal SNN | WeiAttn (Liu et al., 2022) | SDT-V1 | 3.36M | 8 | 0.145 | – | 48.6 |
| | **WeiAttn + PBO** | **SDT-V1** | **3.36M** | **8** | **0.133** | **↓8.3** | **49.1** |
| | SCA (Guo et al., 2023) | SDT-V1 | 3.65M | 8 | 0.162 | – | 48.8 |
| | **SCA + PBO** | **SDT-V1** | **3.65M** | **8** | **0.142** | **↓12.3** | **49.7** |
| | CMCI (Jiang et al., 2023) | SDT-V1 | 4.42M | 8 | 0.174 | – | 48.1 |
| | **CMCI + PBO** | **SDT-V1** | **4.42M** | **8** | **0.155** | **↓10.9** | **49.2** |
| | S-CRML (He et al., 2025) | SDT-V1 | 4.16M | 8 | 0.168 | – | 49.7 |
| | **S-CRML + PBO** | **SDT-V1** | **4.16M** | **8** | **0.147** | **↓12.5** | **51.3** |

the LIF-only baseline remains largely quiescent and under-responsive. By the final stage, PBO maintains sparse yet *functionally engaged* activity patterns, consistent with a well-shaped band-pass. Collectively, these results show that PBO not only improves accuracy substantially but also steers the network toward a more semantically aligned and energetically disciplined operating regime.

**Multi-modal effectiveness with DVS.** Beyond uni-modal action recognition, we evaluate RGB-DVS fusion to further test its effectiveness. PBO is utilized as a drop-in module in the RGB branch, so that it adaptively optimizes the temporal pass-band, which improves complementarity with the DVS stream. Under a unified implementation, attaching PBO to the lightweight spiking fusion model S-CMRL achieves **73.03%**, **74.18%**, **51.30%** accuracy on UCF101-CEP, HMDB51-CEP and HARDVS, respectively, as shown in Table 2 and Table 3. This corresponds to gains of **+4.90**, **+1.85** and **+1.60** percentage points over the baseline without PBO (S-CMRL). It can be seen that the proposed PBO surpasses recent uni-modal methods (TIM, SDT-V2, SDT-V3) and multi-modal SNN fusion methods (WeiAttn, SCA, CMCI). These gains come without modifying backbones or additional parameter budgets, indicating that simple plug-and-play pass-band alignment is sufficient to unlock strong uni-modal and multi-modal improvements. We also provide Class Activation Mapping (CAM) visualizations in Appendix C to better illustrate multi-modal cooperation.

Table 4: Ablation study on UCF101-CEP dataset for leaky factor $\tau$, consistency weight $\alpha$, and modulation amplitude $A$ in $\lambda[t] = \mu + A\sin(\omega t + \phi)$. We report the accuracy (Acc/%).

| Module $\rightarrow$ | Leaky factor $\tau$ | | | | Consistency weight $\alpha$ $(10^{-3})$ | | | | | | Amplitude $A$ | | | |
|---|---|---|---|---|---|---|---|---|---|---|---|---|---|---|
| | 0.3 | 0.5 | 0.7 | 0.9 | 0 | 1 | 5 | 10 | 30 | 50 | 0 | 0.1 | 0.3 | 0.5 |
| Acc | 71.27 | 72.57 | **73.03** | 70.20 | 64.70 | 70.41 | 72.49 | **73.03** | 72.41 | 70.78 | 70.14 | **73.03** | 72.17 | 72.25 |

**Ablation studies.** As shown in Table 4, we ablate the leak factor $\tau$, the consistency weight $\alpha$, and the modulation amplitude $A$. On UCF101-CEP, **the leak factor** $\tau$ exhibits a clear "middle is best" trend: accuracies at $\tau = 0.3/0.5/0.7/0.9$ are $71.27\%/72.57\%/\mathbf{73.03}\%/70.20\%$. Within the $\lambda[t]$ based system, $\tau = 0.7$ induces a relatively strong leaky low-pass that cooperates best with our PBO. Interestingly, across different $\tau$, the learned $\mu$ consistently converges near 1, while the learned $\omega$ varies substantially. Such frequency changes with $\tau$ highlights the necessity of pass-band shifting. Detailed visualizations are provided in Appendix F. **Consistency weight** $\alpha$**:** Removing the term ($\alpha = 0$) reduces accuracy to $64.70\%$, indicating that without the consistency regularizer the optimized pre LIF signal may drift from the original semantics and introduce distortion. Increasing $\alpha$ to $1\times10^{-3}$ and $5\times10^{-3}$ improves accuracy to $70.41\%$ and $72.49\%$. It peaks at $\alpha = 1\times10^{-2}$ with $\mathbf{73.03}\%$, while larger values $(3\times10^{-2}, 5\times10^{-2})$ begin to hurt. **Modulation amplitude** $A$**:** A small $A = 0.1$ already reaches the highest accuracy $73.03\%$. Increasing $A$ to 0.3 and 0.5 yields slight drops but remains stable overall. Removing this term ($A = 0$) degenerates the original time varying system into a time invariant one. As a result, the pass-band cannot be modulated and the accuracy drops sharply to $70.14\%$, further validating our theory and the necessity of pass-band shifting.

## 5.2 Energy Evaluation

We also compare the energy consumption and recognition accuracy of representative ANN, SNN, and multimodal SNN methods on HARDVS, which are summarized in Table 3. Our method attains a favorable balance between performance and efficiency. The measurement protocol and computation details are provided in Appendix B. Compared with uni-modal and multi-modal SNN baselines, PBO reaches $51.3\%$ accuracy with only $0.146\,\text{mJ}$, surpassing all SNN based methods. It is worth noting that it further reduces energy while improving accuracy over all multi-modal fusion methods.

## 5.3 Extension to Video Anomaly Detection

The VAD task requires frame level anomaly scoring over long videos with sparse and irregular events, which makes conventional ANN pipelines energy intensive. To assess scalability under weak supervision, we apply our plug-and-play PBO on the MSF (Qian et al., 2025), implementation details and visualizations are provided in Appendix D. As shown in Table 5, our PBO can effectively improve RGB only VAD method by increas-

Table 5: Results on UCF-Crime and UCF-Crime-DVS.

| Type | Method | Features | AUC(%) | FAR(%) |
|---|---|---|---|---|
| ANN | Sultani et al. (2018) | Event | 55.56 | 8.69 |
| | 3C-Net (Narayan et al., 2019) | Event | 59.22 | 9.50 |
| | AR-Net (Wan et al., 2020) | Event | 60.71 | 8.51 |
| | RTFM (Tian et al., 2021) | Event | 52.67 | 13.19 |
| SNN | MSF (Qian et al., 2025) | Event | 65.01 | **3.27** |
| | | RGB | 71.54 | 14.54 |
| | | RGB + Event | 70.01 | 17.89 |
| | MSF + PBO | RGB | 72.31 | 10.89 |
| | | **RGB + Event** | **74.14** | 5.19 |

ing AUC (Area Under the ROC Curve) and reducing FAR (False Alarm Rate). In the RGB+DVS setting, MSF combined with PBO achieves state-of-the-art performance.

## 6 Conclusion

This paper reframes SNN video understanding as a temporal *pass-band mismatch* and shows that when motion-bearing mid frequencies are prioritized *less* low-frequency content can yield *more* discriminative spikes. We introduce Pass-Band Optimizer (PBO), a plug-and-play, causal prefilter that reshapes the LIF-induced low-pass toward a task-aligned band-pass by suppressing DC/near-DC components and passing midhigh-frequency motion; it adds only two lightweight scalars and requires no backbone changes. Instantiating a time-varying $\lambda[t]$ and a consistency regularizer broadens the optimizable window while preserving semantics, enabling streaming-friendly deployment. Empirically, PBO delivers consistent gains across uni- and multi-modal action recognition on UCF101-CEP, HMDB51-CEP, and HARDVS, and extends to weakly supervised VAD on UCF-Crime-CEP, achieving favorable accuracyenergy trade-offs under the same training budgets. We believe this frequency-oriented view opens avenues for SNN-based video understanding.

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

# A    PASS-BAND CHARACTERISTICS AND THE LIMITATION OF A SINGLE $\lambda$

Using the cascaded magnitude response in Eq. 18, and for readability letting $a = 1 + \lambda^2$, $b = 2\lambda$, $c = 1 + \alpha^2$, $d = 2\alpha$, we can rewrite the pass-band as

$$\left| G(e^{j\omega}, \lambda) \right|^2 = (1-\alpha)^2 \frac{a - b\cos\omega}{c - d\cos\omega}. \tag{29}$$

**Endpoint gains (delimiting the band edges).**

$$\left| G(e^{j0}, \lambda) \right|^2 = (1-\lambda)^2, \qquad \left| G(e^{j\pi}, \lambda) \right|^2 = \frac{(1+\lambda)^2(1-\alpha)^2}{(1+\alpha)^2}. \tag{30}$$

**Tilt vs. flat point (no mid-band peak with a single $\lambda$).**    Differentiating $\frac{a - b\cos\omega}{c - d\cos\omega}$ w.r.t. $u = \cos\omega$ yields

$$\frac{d}{du}\left( \frac{a - bu}{c - du} \right) = \frac{ad - bc}{(c - du)^2}, \qquad ad - bc = 2(\alpha - \lambda)(1 - \alpha\lambda).$$

Hence, for fixed $\alpha$:

$$\lambda < \alpha : \text{ peak at } \omega = 0 \text{ (low-pass tilt);}$$
$$\lambda = \alpha : \left| G \right|^2 \equiv (1-\alpha)^2 \text{ (flat);} \tag{31}$$
$$\lambda > \alpha : \text{ peak at } \omega = \pi \text{ (high-pass tilt).}$$

*Implication.* A single scalar $\lambda$ can only move the passband centroid from low to high frequencies (with a flat point at $\lambda = \alpha$); it cannot create a genuine mid-band peak, *i.e.,* a strict band-pass window.

$-3$ **dB cutoffs (unique solutions).**    Because $|G|^2$ is strictly monotone over $\omega \in [0, \pi]$ when $\lambda \neq \alpha$, each tilt has a unique $-3$ dB cutoff.

**Low-pass tilt** $(\lambda < \alpha)$, normalized at $\omega = 0$:

$$\frac{1 + \lambda^2 - 2\lambda\cos\omega_c^{\text{(LP)}}}{1 + \alpha^2 - 2\alpha\cos\omega_c^{\text{(LP)}}} = \frac{1}{2} \cdot \frac{(1-\lambda)^2}{(1-\alpha)^2}. \tag{32}$$

**High-pass tilt** $(\lambda > \alpha)$, normalized at $\omega = \pi$:

$$\frac{1 + \lambda^2 - 2\lambda\cos\omega_c^{\text{(HP)}}}{1 + \alpha^2 - 2\alpha\cos\omega_c^{\text{(HP)}}} = \frac{1}{2} \cdot \frac{(1+\lambda)^2}{(1+\alpha)^2}. \tag{33}$$

Solving Eq. 32 or Eq. 33 for $\cos\omega_c \in [-1, 1]$ gives the unique cutoff frequency. In practice, $\lambda$ thus controls the passband tilt and edge location under a fixed $\alpha$, but cannot introduce a mid-band bump without extending to a time-varying $\lambda[t]$.

**Limitation of a single $\lambda$.**    A scalar $\lambda$ can only *interpolate* the passband centroid between a low-pass tilt, the flat point, and a high-pass tilt. It *cannot* create a truly *peaked mid-band* (a strict band-pass window), hence the tunable passband shape and center are limited.

# B    ENERGY EVALUATION

Energy consumption is a critical metric for evaluating the performance of SNNs. We estimate the theoretical energy consumption following the methodology in (Yao et al., 2023a). First, the number of synaptic operations (SOPs), which reflect the total number of accumulate (AC) operations triggered by spikes, are estimated as:

$$\text{SOP}_l = R_l \times T \times \text{FLOP}_l, \tag{34}$$

where $R_l \in [0, 1]$ denotes the average spike rate in layer $l$, $T$ is the number of timesteps, and $\text{FLOP}_l$ is the number of floating-point operations in the corresponding non-spiking layer. On 45 nm CMOS

hardware, the energy cost per multiply-accumulate (MAC) operation is $E_{\text{MAC}} = 4.6\,\text{pJ}$, while the cost per accumulate (AC) operation is $E_{\text{AC}} = 0.9\,\text{pJ}$. The total energy consumption is computed as:

$$E_{\text{total}} = E_{\text{MAC}} \times \text{FLOP}_1 + E_{\text{AC}} \times \sum_{l=2}^{L} \text{SOP}_l, \tag{35}$$

where $\text{FLOP}_1$ denotes the number of floating-point operations in the first convolutional layer. For all subsequent layers ($l \geq 2$), spike-driven binary activations are used, and computations are modeled as synaptic operations $\text{SOP}_l$.

## C  VISUALIZATION ANALYSIS

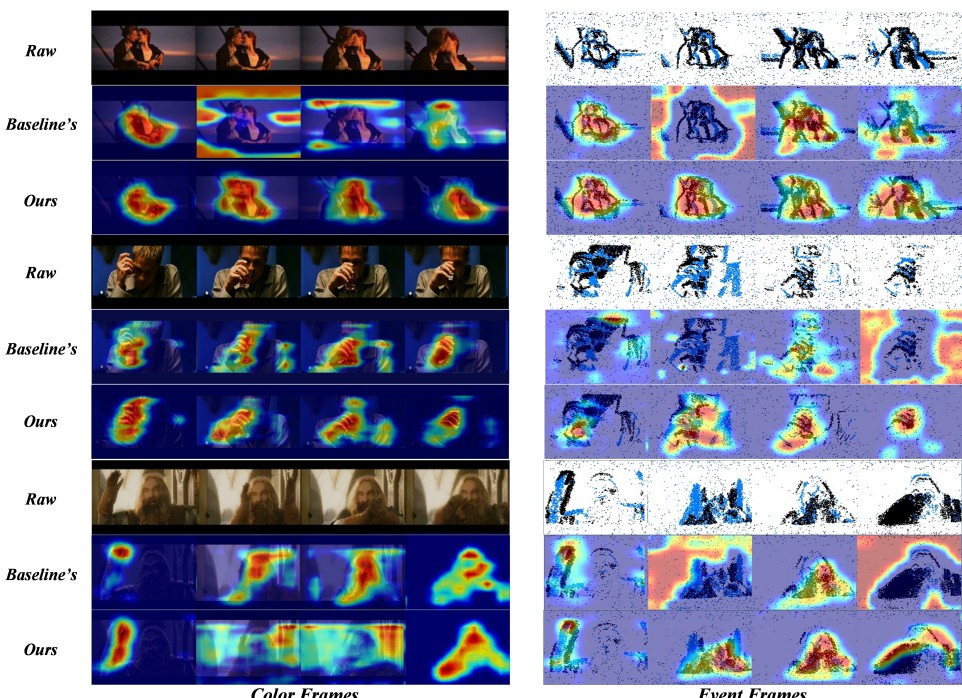

Figure 3: Class Activation Mapping (CAM) on HMDB51-CEP. Three action categories are selected for visualization. Each category includes two columns: color and event frames, and three rows: original inputs, CAM from baseline (S-CRML), and CAM from our method (S-CMRL + PBO).

Fig. 3 presents the Class Activation Map (CAM) visualizations comparing our method with SCA (Guo et al., 2023) across three representative action categories: *kiss*, *drink*, and *clap*. Each class includes two columns (color frames on the left and event frames on the right) and three rows: (1) the raw input frames, (2) CAM visualizations generated by SCA, and (3) visualizations from our method. As illustrated, our approach consistently produces sharper, more focused activation regions that are semantically aligned with the action-relevant parts in both modalities. Benefiting from time-specific sparsity and efficient cross-modal interaction, our model accurately captures the discriminative motion cues while suppressing background noise. In particular, the event (DVS) stream under our method reliably highlights motion-dominant areassuch as the face in *kiss*, the drinking action in *drink*, and the clapping hands in *clap*across all samples. Meanwhile, the RGB modality provides complementary semantic cues, enriching the representation with contextual appearance information. In contrast, the baseline often activates irrelevant or overly broad regions, especially under background clutter or subtle motion. These results demonstrate that our method facilitates more precise spatial-temporal attention, leading to better discriminative feature extraction and cross-modal consistency.

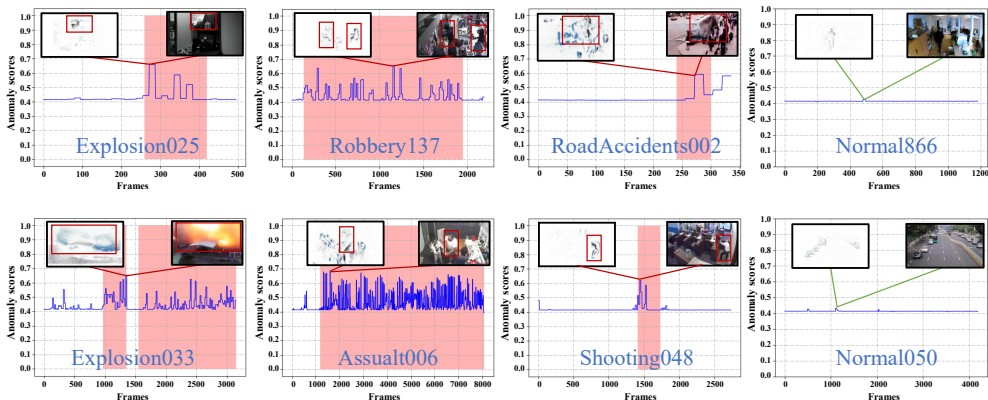

Figure 4: Anomaly scores of our methods on Color-Event UCF-Crime. Pink areas indicate the manually labelled abnormal events, purple lines represent the anomaly score and red boxes point out abnormal events on the screen.

## D  VAD IMPLEMENTATION DETAILS

Following (Sultani et al., 2018; Qian et al., 2025), each video and event stream are divided into 16 non-overlapped clips and the total number of training epochs is set to 20.

### D.0.1  EVALUATION METRICS

Following prior works (Wan et al., 2020; Qian et al., 2025), we report Area Under of Curve (AUC) of the frame-level Receiver Operating Characteristics (ROC) and False Alarm Rate (FAR) with a threshold 0.5. AUC measures the overall discriminative capability of the model, while FAR evaluates its reliability and robustness in real-world scenarios.

### D.0.2  EXPERIMENT RESULTS

As shown in Table 5, our method alone achieves an accuracy of 65.45%, which increases to 71.41% when combined with MSF, outperforming the previous MSF by 0.44% and 6.40%, respectively. This demonstrates that PBO not only sets a new performance benchmark for SNNs in weakly supervised video anomaly detection, but also that the integration with MSF highlights the effectiveness of our PBO in providing temporally and semantically coherent features that are more amenable to SNN learning. These results validate our method both as a feature enhancement module and a viable backbone, marking a significant step toward closing the performance gap between SNNs and ANNs in this domain.

### D.1  VISUALIZATION

Fig. 4 presents a set of visualizations demonstrating that PBO effectively distinguishes normal from abnormal events. For instance, in Shooting048, the anomaly scores rise sharply when individuals raise and fire guns. In Robbery137, although the anomaly scores do not consistently exceed the threshold throughout the anomalous segments, this is attributed to relatively static scenes that fail to trigger event responses in the DVS, resulting in partial information loss. Nevertheless, elevated scores are observed during key moments, such as gun possession and the act of stealing from cabinets. For explosion events, which share visual patterns with scene transitions or light flickering in DVS data, PBO is able to differentiate them accurately, thereby reducing false alarms.

## E  WHY LTV IS REASONABLE?

**Approximation to a constant-$\lambda$ filter.**  Let $\lambda[t] = \mu + \delta[t]$ with time average $\frac{1}{T}\sum_{t=0}^{T-1}\delta[t] \to 0$ and variance $\frac{1}{T}\sum_{t=0}^{T-1}\delta[t]^2 \to \sigma_\lambda^2$. The instantaneous frequency response of the two-tap pre-filter is

$$H_t(e^{j\omega}) = 1 - \big(\mu + \delta[t]\big)e^{-j\omega}. \tag{36}$$

Averaging the squared gain over time yields

$$\overline{|H|^2}(\omega) \triangleq \lim_{T \to \infty} \frac{1}{T} \sum_{t=0}^{T-1} \left| H_t(e^{j\omega}) \right|^2 = \left| 1 - \mu e^{-j\omega} \right|^2 + \lim_{T \to \infty} \frac{1}{T} \sum_{t=0}^{T-1} \delta[t]^2, \tag{37}$$

because the cross-term vanishes by the zero-mean assumption on $\delta[t]$. Hence

$$\overline{|H|^2}(\omega) = \underbrace{\left| 1 - \mu e^{-j\omega} \right|^2}_{\text{constant-}\lambda \text{ template}} + \sigma_\lambda^2 = \left( 1 + \mu^2 - 2\mu \cos\omega \right) + \sigma_\lambda^2. \tag{38}$$

Therefore, when the variance $\sigma_\lambda^2$ is small (or treated as an $\omega$-independent offset), the time-varying filter $\lambda[t]$ is well approximated by the constant-$\lambda$ filter with $\lambda = \mu$ in the sense of average squared gain.

**High-pass property preserved at the mean.** The constant-$\lambda$ template satisfies

$$\left| 1 - \mu e^{-j\omega} \right|^2 = 1 + \mu^2 - 2\mu \cos\omega \approx (1 - \mu)^2 + \mu \omega^2 \quad (\omega \to 0), \tag{39}$$

so $\omega = 0$ is a local minimum for any $\mu > 0$, *i.e.,* the response is high-pass around DC. Since $\sigma_\lambda^2$ in Eq. 38 is $\omega$-independent, the high-pass shape (low-frequency suppression) is preserved under the approximation $\lambda[t] \approx \mu$.

# F   VISUALIZATION OF $\mu$ AND $\omega$ IN $\lambda[t]$

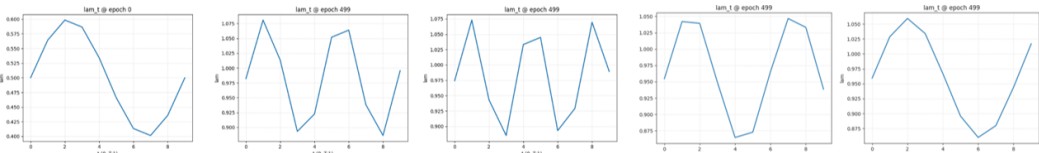

Figure 5: **Visualization of the learned temporal mixing weights** $\lambda[t]$ under different leak factors $\tau$. From left to right: initialization (epoch 0), and results at epoch 499 for $\tau = 0.3$, 0.5, 0.7, 0.9.
**Analysis of the modulation patterns in Fig. 5.** Figure 5 plots the learned temporal mixing weights $\lambda[t]$ over $T = 10$ steps. The leftmost panel shows the initialization (epoch 0), which is a gentle single-period sinusoid. After training (epoch 499), the four panels correspond to $\tau = 0.3$, 0.5, 0.7, 0.9, respectively. As $\tau$ increases, $\lambda[t]$ exhibits progressively higher temporal frequencyroughly from $\sim 1$ period at initialization to $\gtrsim 2$ periods when $\tau$ is largewhile remaining centered near $\mu \approx 1$ with a small amplitude ($A \approx 0.05$–$0.1$). This behavior is consistent with the LIF update

$$U[t] = (1 - \tau) V[t-1] + \tau X[t] + \text{const},$$

which yields a first-order low-pass whose memory shortens as $\tau$ grows. To complement this shorter memory and the wider pass-band of the LIF cell, the modulator increases its switching rate (higher $\omega$), i.e., it mixes the DC and differential references more rapidly within the same window.

# G   FULL PSD DERIVATION WITHOUT THE UNCORRELATED-FREQUENCY ASSUMPTION

This section provides the complete derivation of the output power spectral density (PSD) when frequency components of the input signal may be correlated. No decorrelation assumption is used here. The result shows that the translated sidebands induced by temporal modulation remain present regardless of the correlation structure.

## G.1   SPECTRUM OF THE CASCADED LIF–MODULATION SYSTEM

From Eq. equation 9, the output spectrum under a general harmonic modulation is

$$Y(e^{j\omega}) = H_{\text{LIF}}(e^{j\omega}) \sum_{m \in \mathbb{Z}} W_m(e^{j\omega}) X\left( e^{j(\omega - m\omega_0)} \right). \tag{40}$$

The corresponding PSD is

$$S_{\text{out}}(\omega) = \mathbb{E}\left[ \left| Y(e^{j\omega}) \right|^2 \right]. \tag{41}$$

## G.2 EXPANSION OF THE PSD

Substituting Eq. equation 40 into the above yields

$$S_{\text{out}}(\omega) = \left|H_{\text{LIF}}(e^{j\omega})\right|^2 \sum_m \sum_n W_m(e^{j\omega})W_n^*(e^{j\omega})\,\mathbb{E}\Big[X\Big(e^{j(\omega-m\omega_0)}\Big)X^*\Big(e^{j(\omega-n\omega_0)}\Big)\Big]. \tag{42}$$

Define the generalized cross-spectrum

$$S_X(\alpha,\beta) \;=\; \mathbb{E}\Big[X(e^{j\alpha})\,X^*(e^{j\beta})\Big], \tag{43}$$

then Eq. equation 42 becomes

$$S_{\text{out}}(\omega) = \left|H_{\text{LIF}}(e^{j\omega})\right|^2 \sum_m \sum_n W_m(e^{j\omega})W_n^*(e^{j\omega})S_X(\omega-m\omega_0,\,\omega-n\omega_0). \tag{44}$$

This is the full PSD without approximations.

## G.3 DECOMPOSITION INTO DIAGONAL AND CROSS-SPECTRAL COMPONENTS

The terms with $m = n$ correspond to the auto-spectral contributions:

$$S_X(\omega-m\omega_0,\,\omega-m\omega_0) = S_{\text{in}}(\omega-m\omega_0). \tag{45}$$

The remaining terms with $m \neq n$ collect all cross-spectral correlations:

$$S_X(\omega-m\omega_0,\,\omega-n\omega_0), \qquad m \neq n. \tag{46}$$

Thus, the PSD may be written as

$$S_{\text{out}}(\omega) = \left|H_{\text{LIF}}(e^{j\omega})\right|^2 \Bigg[\underbrace{\sum_m \left|W_m(e^{j\omega})\right|^2 S_{\text{in}}(\omega-m\omega_0)}_{\text{auto-spectral terms}}$$

$$+ \underbrace{\sum_{m\neq n} W_m(e^{j\omega})W_n^*(e^{j\omega})\,S_X(\omega-m\omega_0,\,\omega-n\omega_0)}_{\text{cross-spectral terms}}\Bigg]. \tag{47}$$

## G.4 SINGLE-TONE MODULATION CASE

For the single-tone modulation used in the main paper, only $m \in \{0,\pm 1\}$ are non-zero. Let

$$X_0 = X(\omega), \quad X_+ = X(\omega-\omega_0), \quad X_- = X(\omega+\omega_0),$$

and similarly for $W_0, W_+, W_-$. Substituting into Eq. equation 47 yields

$$S_{\text{out}}(\omega) = \left|H_{\text{LIF}}(e^{j\omega})\right|^2 \Big[|W_0|^2 S_{\text{in}}(\omega) + |W_+|^2 S_{\text{in}}(\omega-\omega_0) + |W_-|^2 S_{\text{in}}(\omega+\omega_0)$$

$$+ 2\Re\Big(W_0 W_+^* \, S_X(\omega,\omega-\omega_0) + W_0 W_-^* \, S_X(\omega,\omega+\omega_0) + W_+ W_-^* \, S_X(\omega-\omega_0,\omega+\omega_0)\Big)\Big]. \tag{48}$$

## G.5 IMPLICATION FOR THE LEARNED PASS-BAND

The terms

$$|W_+|^2 S_{\text{in}}(\omega-\omega_0), \qquad |W_-|^2 S_{\text{in}}(\omega+\omega_0),$$

correspond to the frequency-translated sidebands characteristic of harmonic modulation. These terms remain present irrespective of the cross-spectral correlations in the input signal. The correlation-dependent expressions

$$S_X(\omega,\omega-\omega_0), \quad S_X(\omega,\omega+\omega_0), \quad S_X(\omega-\omega_0,\omega+\omega_0)$$

modify amplitudes but cannot cancel the translated components. Consequently, the learned modulation continues to produce a nonzero mid-band emphasis even in the fully correlated case.

Table 6: Ablation on clip length $T$ on UCF101-CEP (stride = 1).

| $T$ | Stride | PBO | Top-1 Acc (%) |
|---|---|---|---|
| 4 | 1 | w/o | 49.33 |
| 4 | 1 | w/ | 56.67 |
| 8 | 1 | w/o | 66.65 |
| 8 | 1 | w/ | 72.60 |
| 10 | 1 | w/o | 68.13 |
| 10 | 1 | w/ | 73.03 |
| 16 | 1 | w/o | 59.95 |
| 16 | 1 | w/ | 67.53 |

Table 7: Ablation on temporal sampling stride on UCF101-CEP ($T = 8$).

| $T$ | Stride | PBO | Top-1 Acc (%) |
|---|---|---|---|
| 8 | 1 | w/o | 66.65 |
| 8 | 1 | w/ | 72.60 |
| 8 | 2 | w/o | 66.22 |
| 8 | 2 | w/ | 71.26 |
| 8 | 4 | w/o | 65.90 |
| 8 | 4 | w/ | 70.81 |

Table 8: Ablation on input spatial resolution on UCF101-CEP ($T = 10$, stride = 1).

| Resolution | PBO | Top-1 Acc (%) |
|---|---|---|
| 32 | w/o | 59.68 |
| 32 | w/ | 67.74 |
| 64 | w/o | 68.13 |
| 64 | w/ | 73.03 |
| 128 | w/o | 65.82 |
| 128 | w/ | 72.87 |

Table 9: Ablation on consistency-loss components on UCF101-CEP.

| ID | $L_{\text{int}}$ | $L_{\text{grad}}$ | Description | Top-1 Acc (%) |
|---|---|---|---|---|
| S1 | ✗ | ✗ | No consistency loss | 64.70 |
| S2 | ✓ | ✗ | $L_{\text{int}}$ only | 72.17 |
| S3 | ✗ | ✓ | $L_{\text{grad}}$ only | 69.64 |
| S4 | ✓ | ✓ | Full consistency loss | 73.03 |

Table 10: Ablation on HMDB-CEP for leaky factor $\tau$, consistency weight $\alpha$, and modulation amplitude $A$ in $\lambda[t] = \mu + A\sin(\omega t + \phi)$. We report Top-1 accuracy (%).

| Module → | Leaky factor $\tau$ | | | | Consistency weight $\alpha$ ($10^{-3}$) | | | | | | Amplitude $A$ | | | |
|---|---|---|---|---|---|---|---|---|---|---|---|---|---|---|
| | 0.3 | 0.5 | 0.7 | 0.9 | 0 | 1 | 5 | 10 | 30 | 50 | 0 | 0.1 | 0.3 | 0.5 |
| Acc (%) | 73.43 | 73.58 | **74.18** | 73.13 | 71.94 | 72.99 | 73.43 | **74.18** | 73.58 | 72.24 | 71.64 | **74.18** | 74.18 | 72.98 |

## H  MORE EXPERIMENTS

This section provides additional experimental results referenced in the main paper, together with extended ablations for a more complete analysis of the proposed PBO module. We report results on: (1) clip length, (2) temporal sampling stride, (3) input spatial resolution, (4) the role of the consistency-loss components, (5) hyperparameter sensitivity of the leaky factor $\tau$, consistency weight $\alpha$, and modulation amplitude $A$, (6) comparison between global and per-channel PBO parameterization, and (7) additional evaluations on two modern SNN backbones (QKFormer and SVFormer). Unless otherwise specified, all experiments follow the same training protocol and implementation details as in the main paper.

Table 11: Effect of PBO on additional SNN backbones (UCF101-CEP).

| Backbone | PBO | Top-1 Acc (%) | Δ (%) | Comment |
|---|---|---|---|---|
| QKFormer4-384 | w/o | 45.20 | – | Baseline |
| QKFormer4-384 | w/ | **54.62** | +9.42 | Clear improvement |
| SVFormer-base | w/o | 63.55 | – | Baseline |
| SVFormer-base | w/ | **69.37** | +5.82 | Strong gain |

## USE OF LARGE LANGUAGE MODELS (LLMS)

We used a large language model (LLM) solely as a writing assistant for *language editing* grammar correction, wording/fluency polishing, and minor rephrasing for clarity and for *retrieval and discovery* to surface potentially relevant related work and references. The LLM was *not* involved in research ideation, problem formulation, methodology or experiment design, coding, data analysis, result generation, or drawing conclusions. All candidate references returned by the LLM were screened and selected by the authors; all technical content and conclusions were authored and verified by the human authors, who take full responsibility for the paper. The LLM is not eligible for authorship. Further details of these uses are described in the paper.

## ETHICS STATEMENT

This work proposes a plug-and-play pass-band optimizer for spiking neural networks (SNNs) for video action recognition and weakly supervised video anomaly detection. We evaluate only on public benchmarks with paired RGB and event (DVS) streams UCF101/UCF101-DVS (aligned as UCF101-CEP), HMDB51/HMDB51-DVS (HMDB51-CEP), HARDVS, and UCF-Crime/UCF-Crime-DVS; no new data were collected, and no human subjects or personally identifiable information are involved. Aware of dual-use risks in video understanding (*e.g.,* surveillance), we restrict our study to public datasets, release any artifacts for research-only use, provide no deployment-oriented functionality, and do not target identification or re-identification. For reproducibility and environmental responsibility, we report implementation details, keep compute modest (training on four NVIDIA RTX 4090 GPUs), and estimate energy using standard SNN synaptic-operation accounting with established CMOS energy costs, reporting accuracy–energy trade-offs and encouraging license-compliant, responsible use.

