# OpenReview forum: "Fire on Motion: Optimizing Video Pass-bands for Efficient Spiking Action Recognition"
_ICLR.cc/2026/Conference — Submitted to ICLR 2026_

### Official Review · Reviewer_iX1W · 2025-10-27

**Soundness:** 3
**Presentation:** 3
**Contribution:** 2
**Rating:** 6
**Confidence:** 3

**Summary:**

This paper identifies a key limitation of spiking neural networks (SNNs) for video understanding, a temporal pass-band mismatch where the Leaky Integrate-and-Fire (LIF) neurons act as low-pass filters that suppress motion-bearing mid-frequency information crucial for dynamic tasks. To address this, the authors propose the Pass-Band Optimizer (PBO), a plug-and-play causal pre-filter with only two learnable parameters that reshapes the LIF’s effective temporal response from low-pass to task-aligned band-pass. By introducing a time-varying coefficient and a lightweight consistency constraint, PBO adaptively emphasizes motion frequencies while preserving semantic fidelity, requiring no architectural change or extra cost. Experiments on multiple video datasets (UCF101, HMDB51, HARDVS, UCF-Crime) show consistent accuracy improvements (up to +11%) and reduced energy usage.

**Strengths:**

(1) The analysis of LIF dynamics as a low-pass filter (Eq. 9–13) and the decomposition of input spectra (Eq. 6–11) are well-grounded and insightful.

(2) Results across uni-modal and multi-modal (RGB + DVS) benchmarks demonstrate strong consistency and good energy–accuracy trade-off.

**Weaknesses:**

(1) The method assumes that a single learned pass-band generalizes across all videos, even though motion frequencies differ widely between actions, which raises concerns about adaptability and robustness.

(2) The approach does not explicitly define or analyze the cutoff frequency of the learned pass-band, making it hard to interpret or control how PBO shapes the spectral response.

(3) The choice of time constant τ = 0.7 is empirically driven and not theoretically justified, even though τ strongly affects the low-pass characteristics of the LIF neuron and therefore interacts with the learned parameters μ and ω.

(4) The paper presents Figure 1 showing temporal power spectra of UCF101 videos and different filtering effects. However, it is unclear how this figure was generated given that UCF101 contains a large number of diverse clips. Was the spectrum computed from a single representative video, a subset, or averaged over the entire dataset? If it is an aggregate, how were temporal frequencies aligned and normalized across samples of varying length and motion statistics? Clarifying this would help interpret how general the observed pass-band mismatch is and whether the illustrated spectrum truly reflects dataset-level behavior or just a small subset of examples.

Overall, while the method is elegant and effective, its spectral behavior and parameter dependencies remain somewhat opaque, suggesting room for deeper analysis or joint optimization of τ and the PBO parameters.

**Questions:**

Please address questions in Weakness.

---

### Official Review · Reviewer_8Big · 2025-10-29

**Soundness:** 3
**Presentation:** 3
**Contribution:** 3
**Rating:** 6
**Confidence:** 4

**Summary:**

This paper diagnoses a pass-band mismatch in Spiking Neural Networks (SNNs), where standard LIF neurons act as low-pass filters, suppressing crucial motion-bearing frequencies. The authors propose the Pass-Band Optimizer (PBO), a lightweight, plug-and-play pre-filter that adds only two learnable parameters to high-pass the input stream and focus on motion. A consistency loss is used to preserve semantics. The method achieves significant gains, such as +11.55% on UCF101.

**Strengths:**

1. Clear problem statement
- The paper’s primary strength is its clear, frequency-domain analysis of the "pass-band mismatch" in SNNs. Reframing the SNN video processing bottleneck from a signal-processing perspective is a good contribution that clearly explains why SNNs, despite their temporal nature, might fail at motion tasks.

2. Extensive experiments
- The method achieves substantial, not marginal, accuracy gains such as +10.55% and +11.55% on UCF101.

3.  Efficiency
- PBO is a lightweight (2 parameters), plug-and-play module with negligible overhead, making it highly practical.

4. Generalizability
- This work is validated across multiple backbones, datasets, and three distinct task types (uni-modal, multi-modal, VAD) .

**Weaknesses:**

1. Pass-Band claims
- Figure 1(c) says that PBO creates 'task-optimal pass-band', but I cannot find other cases.

2. Limited Ablation studies
- Loss combination & only applied to UCV101-CEP dataset.

3. Hyperparameter setting
- Related to the second weakness. Are A=0.1 and $\alpha=1\times 10^-2$ optimal parameters for every dataset?

4. Theoretical inconsistency
- In Appendix A, the authors said a single $\lambda$ cannot create mid-band peak. However, Appendix E shows $\lambda [t] \approx$ constant. This contradiction makes me confused.

5. The text of the figures is too tiny.

**Questions:**

1. Can authors provide the ablation study regarding the loss components? (only $L^{int}$ or $L^{grad}$)

2. I doubt the optimal hyperparameters in Table 4. Can you provide ablation studies with other dataset for generalization?

3. The authors mentioned that PBO can create a task-optimal pass band. But only one example exists. I just wondered if we can check the actual frequency response for different tasks or datasets.

4. Based on the experiment tables, the authors applied PBO on SDT and Spikformer. Can authors apply PBO to other architecture, such as QKFormer, for generalization?

---

### Official Review · Reviewer_dncK · 2025-11-01

**Soundness:** 2
**Presentation:** 2
**Contribution:** 2
**Rating:** 4
**Confidence:** 4

**Summary:**

The paper diagnoses a temporal pass‑band mismatch in SNNs: the LIF membrane behaves as a temporal low‑pass so near‑DC content passes while motion‑bearing mid‑frequencies are attenuated. The authors analyze this via the LIF frequency response and propose a plug‑and‑play, two‑tap, linear time‑varying pre‑filter placed before the first spiking membrane. Learning $\mu$ and $\omega$ adapts the effective pass‑band, and a consistency loss keeps semantics and boundaries stable. The module aims for negligible overhead and no architecture changes.

**Strengths:**

1. Motivation is clear and backed by analysis. The LIF low‑pass characterization is explicit (Eq. 9 with Fig. 1).
2. Small, deployable change. The pre‑filter is two‑tap, streaming‑friendly, and sits outside the backbone. The claim of “two learnable parameters” is attractive for deployment.
3. Experiments show consistent empirical gains across tasks/backbones without architectural edits.

**Weaknesses:**

1. The paper repeatedly emphasizes only two learnable scalars ($\mu,\omega$), yet Table 4 ablates amplitude $A$, which materially changes accuracy. Clarify whether $A$ (and $\phi$) are fixed or tuned per dataset—this affects both training efficiency and fair comparison claims. If $A$ is tuned per dataset, the comparison is not purely “two‑scalar” anymore; please make this precise.
2. A fair question is whether *simpler* band‑pass designs (e.g., a time-invariant learnable) would match PBO. The paper only contrasts “PBO+LIF” and “LIF‑only” (Fig. 2) rather than a richer set of fixed band‑pass controls.
3. L50-53: "However, pure first-order differencing removes DC entirely..." DC was referenced before definition.

**Questions:**

1. How sensitive is performance to clip length $T$, sampling stride, and input resolution?
2. Could a per‑channel $\mu,\omega$ (or per‑stage) version offer further gains with tiny extra cost?
3. Any preliminary results on larger/modern video sets (e.g., Kinetics or Something‑Somethingv2) to check scalability of the pass‑band idea? The temporal-related dataset SSv2 can make the results more convincing. Of course, this is only a friendly suggestion given the short rebuttal period.

---

### Official Review · Reviewer_6T5D · 2025-11-02

**Soundness:** 1
**Presentation:** 2
**Contribution:** 2
**Rating:** 2
**Confidence:** 5

**Summary:**

This paper proposed a Pass-Bands Optimizer to optimize the temporal pass-band toward task-relevant motion bands.

**Strengths:**

The proposed Pass-Bands Optimizer improves performance on various tasks.

**Weaknesses:**

1.	The theoretical derivation in this article is inaccurate:

（1）	From Eqs. (2–5), the authors try to link the time domain convolution in Eq. (2) to a band pass filter; However, $\omega_1<\omega_2$ is not ensured, and Eqs. (3-5) are disconnected from Eq. (2). The claimed band-pass property is ungrounded.

（2）	Eqs (7-9) are largely overlap with [1], merely rewriting the same expressions from the Z domain into the frequency. In line 172, the authors state that $|H_{LIF}(e^{j0})|^2 = 1$; however, the actual response of an LIF neuron is $|H_{LIF}(j\omega)| = 1/\sqrt{(1+(ωτ)^2)}$, with a cut-off frequency of $1/τ$. Therefore, contrary to the authors’ claim, the ultra-low frequency components are in fact attenuated rather than fully passed.

（3）	In Eqs. (11-14), the nonlinear processes of LIF are not discussed; therefore, the discussion of frequency domain characteristics is limited to a single LIF layer, rather than its transmission characteristics within the network.

（4）	In Eq. (24), the approximation of “uncorrelated frequency bins” in video signals is physically and statistically unfounded. The spectral components in video signals, e.g., the energy of $\omega$ and $ω-mω_0$ is instead strongly correlated. Consequently, the “mid band pass window” derived in Eq. (25) cannot be considered theoretically reliable as the correlated (cross-spectral) terms are ignored.

[1] Spiking Transformers need high-frequency information. 2025.

2.	Experiment: Problems exist in the experimental design.

(1) Figure 1 is hard to read. In Fig. 1(a), the high-frequency attenuation appears defined as the gap between the raw and LIF spectral centroids. However, this gap in the high pass case (Fig. 1b) is narrower than in the proposed method (Fig. 1c). So, what exactly is the author trying to convey? Is it to reduce the frequency gap in the high-frequency region, or to keep the frequency gap constant across the entire frequency band? Furthermore, how did the authors arrive at the conclusion that “raw videos concentrate energy at DC, whereas task-relevant motion lies in the mid bands”? Fig. 1(a) does not appear to offer any further discussion of the spectral characteristics of task-relevant motion.

(2) This article consistently emphasizes optimization of the mid-band information. However, the paper does not provide direct analysis to illustrate the correlation between mid-band information and performance. Main comparisons in Table 1-2 are mainly self-implemented, and their reliability cannot be evaluated.

(3) The implementation of PBO actually adds synaptic activity during membrane potential update, which is not taken into account in the energy consumption estimation of the method provided in Appendix B. This makes comparisons of energy consumption in experiments (PBO vs. non-PBO) unfair.

3.	There are several logical inconsistencies in the writing. For example:

（1）	The abstract first attributes the poor dynamic performance of SNNs to the field's overemphasis on static tasks, then claims that the underlying issue is instead due to the “pass-band mismatch”.

（2）	It is a well-known fact that SNNs perform better on neuromorphic (event-driven) datasets, which should be classified as “dynamic input”. However, the authors point out on line 68 that previous studies only dealt with static data, and then acknowledge on lines 116-117 that previous studies have also validated on DVS(event) tasks. DVS recognition is not dominated by static appearance.

（3）	The statement that “since there is little analysis of SNNs, we perform analysis and optimization in the frequency domain” is logically strange. What is the author’s motivation? Why choose the frequency domain as the optimization space? Why not choose other transform domains? For the analysis of LTI systems, [1] has already provided a lot of information.

**Questions:**

See Weakness. Besides, in Figure 3, the CAM visualization of baseline model seem to indicate that the CAM is not set up properly. Could the author please explain? Furthermore, how can it be proven that changing parameters such as λ can create an arbitrarily adjustable mid-band window?

---

### Meta-Review · Area_Chair_L51z · 2025-12-29

**Summary:**

The AC recommends rejection because, without an author rebuttal, the consensus of reviews leans toward rejection with serious theoretical + methodological concerns left unanswered. Reviewer 6T5D’s detailed critic raises doubts about the correctness of the frequency‑domain derivation and the validity of key assumptions underpinning PBO’s “task‑optimal mid‑band” justification, and also questions the fairness of the energy comparisons and the clarity of the experimental figures. Reviewer dncK and Reviewer 8Big see the idea as promising and the empirical gains as substantial, but both call for additional ablations, clearer parameterization (especially regarding amplitudes and loss components), and more thorough spectral characterization across tasks to support the strong claims about pass‑band shaping. Reviewer iX1W considers the analysis insightful but emphasizes that the learned spectral behavior and parameter dependencies remain opaque, particularly given the fixed \tau and the assumption of a single pass‑band. With two reviewers only marginally positive and explicitly open to rejection, one marginally negative, and one strongly negative with high confidence on theoretical issues, and with no rebuttal to clarify or rectify these problems, the AC does not see sufficient grounds to recommend acceptance.
The AC highly recommends the authors to address the concerns of the reviewers and take into account their suggestions of improvement when preparing a revised version.

**Reviewer Concerns:**

Because there was no author response, no reviewer concerns have been explicitly addressed.

**Reviewer Scores:**

The reviewers recommend 2, 4, 6, 6 ratings, the AC recommends rejection. There is no rebuttal/author response provided.

---

### Decision · Program_Chairs · 2026-01-26

Reject